

# Multi-point galactic cosmic rays measurements between 1 and 4.5 AU over a full Solar cycle

Thomas Honig[1], Olivier G. Witasse[1], Hugh Evans[1], Petteri Nieminen[1], Erik Kuulkers[1], Matt G.G.T. Taylor[1], Bernd Heber[2], Jingnan Guo[2], Beatriz Sánchez-Cano[3]

[1] European Space Agency, ESTEC, Noordwijk, 2200 AG, The Netherlands
[2] Institute of Experimental and Applied Physics, Christian-Albrechts-Universitycity, Kiel, Germany
[3] Department of Physics and Astronomy, University of Leicester, Leicester, United Kingdom

*Correspondence to*: Olivier Witasse (owitasse@cosmos.esa.int)

**Abstract.** The radiation data collected by the Standard Radiation Environment Monitor (SREM) aboard ESA missions INTEGRAl, ROSETTA, HERSCHEL, PLANCK and PROBA-1, and by the High Energy Neutron Detector (HEND) instrument aboard Mars Odyssey are analysed with an emphasis on characterising Galactic Cosmic Rays (GCRs) in the inner heliosphere. A cross-calibration between all sensors was performed for this study, which can also be used in subsequent works. We investigate the stability of the SREM detectors over long-term periods. The radiation data is compared qualitatively and

quantitatively with the corresponding solar activity. Based on INTEGRAL and Rosetta SREM data, a GCR helioradial gradient of 2.96%/AU is found between 1 and 4.5 AU. In addition, the data during the last phase of the Rosetta mission around comet 67P/Churyumov-Gerasimenko were studied in more detail. An unexpected and yet unexplained 8% reduction of the Galactic Comic Ray flux measured by Rosetta SREM in the vicinity of the comet is noted.

## 1 Introduction

The space radiation environment affects both manned and unmanned missions outside the Earth's protecting atmosphere and its magnetic field. Highly energetic particles can penetrate living tissue and spacecraft's component material causing damage due to the deposition of energy. Major sources of this radiation are the Van Allen radiation belts, Solar Energetic Particles (SEPs) and Galactic Cosmic Rays (GCRs). This work focusses on the third source, the GCRs, and in particular on their variations in the inner heliosphere. The variation of GCRs as a function of different factors (solar cycle, heliocentric distance,

solar wind conditions) is an interesting topic to explore, and lead to a better understanding of the heliosphere (e.g. Heber and Potgieter, 2008; Heber et al., 2013; Lawrence et al., 2016). The study of the effects of GCRs on the Earth's atmosphere and climate is also a fascinating field of research (e.g Carslaw et al., 2002; Pierce 2017, Everton et al., 2018).

This work is based on the analysis of data collected by the Standard Radiation Environment Monitor (SREM) units on Rosetta,

Integral, Herschel, Planck and Proba-1 spacecraft and on data from the High Energy Neutron Detector (HEND) onboard Mars



Odyssey. While Integral, Herschel, Planck and Proba-1 are located at around 1 AU from the Sun and HEND is orbiting Mars with an average heliocentric distance of 1.5 AU, Rosetta's heliocentric distance varied from 1 to 4.5 AU during its mission lifetime. This combined dataset provides a unique opportunity to determine the GCR flux measured over a range of heliocentric distances up to 3.5 AU and a time period of more than one solar cycle in interplanetary space. Of special interest are the Rosetta

measurements close to comet 67P/Churyumov-Gerasimenko.

## 2 Instrument descriptions and data sets

### 2.1 The ESA radiation monitors

The SREM [e.g. Evans et al., 2008] is a particle detector developed to provide radiation information on a broad range of ESA space missions. SREM instruments have been installed on seven spacecraft so far, with two of them still operating at the time

of writing. With its ability to measure high energetic charged particles (e.g. electrons and protons), it is able to provide valuable information regarding the near platform radiation environment, on short and long terms. In addition, measurements are also a valuable resource for the improvement of space radiation environment models.

The SREM instrument consists of two detector heads with three silicon diode detectors, denoted as D1, D2 and D3. In the first

of the two detector heads, the detectors D1 and D2 are arranged in a telescope configuration with the main entrance covered by 2 mm of aluminium that provides a lower energy threshold of about 20 MeV for protons and about 1.5 MeV for electrons [Mohammadzadeh et al., 2003]. Additionally, the detectors are separated from each other by another 1.7 mm of aluminium and 0.7 mm of tantalum, which sets the threshold for protons up to roughly 39 MeV. Therefore, coincidence of D1 and D2 measures mostly high energetic protons. The opening window for the remaining detector head corresponding to detector D3

is covered with 0.7 mm of aluminium and provides therefore an energy threshold of about 0.5 MeV for electrons and about 10 MeV for protons, respectively. The opening angle of the telescope is ±20 degrees. The detector electronics can operate with a detection rate up to 100 kHz with a corresponding dead-time correction below 20%. The instrument itself is a box of 20x12x10 cm$^3$ which weighs 2.6 kg including the detector system and the supporting electronics. Measuring the incident radiation, the particle events are binned into 15 different channels which have different energy thresholds and discriminator levels. This

allows a differentiated insight into the energy ranges of the events. Table 1 displays the channels with corresponding logic, particles species, energy thresholds and discriminator levels. Channels TC1, S12, S13 (all D1) and TC2 (D2) are sensitive to both electrons and protons, where TC2 has the highest energy threshold of about 49 MeV for protons and about 2.8 MeV for electrons. With the channels S14, S15, C1-C3, S33 and S34 it is possible to measure mainly protons due to the given energy thresholds and the comparatively high discriminator levels. Channel S25 is dedicated to measure the generally very low heavy

ion flux due to its very high discriminator level. However, previous studies point to the fact that the heavy ion channel is most sensitive to protons [Ludecke et al., 2017]. The coincidence channels C1 to C4 use both detector D1 and D2 simultaneously and measure mainly protons due to the high shielding provided by the layers made of aluminium and tantalum. The insensitivity





of the C1, C2 and C3 channels to electrons arises from the high energy deposit thresholds for these channels. The threshold for C4 is low enough to detect the electrons that can make it through the shielding. Channels TC3 and S32 to S34, based on detector D3, are sensitive to low energy protons with the sensitivity to electrons diminishing from S32 to S34. Nevertheless, one should keep in mind that all channels measure electrons as well as protons and that all channels are correlated. This means

5   that it is possible to measure the same event in multiple channels. While the single detector channels tend to measure particles in an omnidirectional way, the coincidence channels can be characterized to measure particles with a certain directionality. Therefore, there is a reduced number of degrees of freedom since the particles are required to deposit energy in D1 and D2 simultaneously, and this is only possible if the particle trajectory crosses both detectors.

| Bin | Logic | Discriminator Level ΔE>XX [MeV] | Proton Energy [MeV] | | | Electron Energy [MeV] | | |
|---|---|---|---|---|---|---|---|---|
| | | | Min [MeV] | Max [MeV] | SCF [#/cm²] | Min [MeV] | Max [MeV] | SCF [#/cm²] |
| TC1 | D1 | 0.085 | 27 | ∞ | 15.8 | 2.0 | ∞ | 118 |
| S12 | D1 | 0.25 | 26 | ∞ | 19.0 | 2.08 | ∞ | 195 |
| S13 | D1 | 0.6 | 27 | ∞ | 16.0 | 2.23 | ∞ | 519 |
| S14 | D1 | 2.0 | 24 | 542 | 38.5 | 3.2 | ∞ | 25403 |
| S15 | D1 | 3.0 | 23 | 434 | 65.6 | 8.18 | ∞ | 5460 |
| TC2 | D1 | 0.085 | 49 | ∞ | 13.1 | 2.8 | ∞ | 191 |
| S25 | D1 | 9.0 | 48 | 270 | 208.8 | n/a | n/a | |
| C1 | D1×D2 | 0.6, 2.0 | 43 | 86 | 107.22 | n/a | n/a | |
| C2 | D1×D2 | 0.6, 1.1 - 2.0 | 52 | 278 | 75.6 | n/a | n/a | |
| C3 | D1×D2 | 0.6, 0.6 - 1.1 | 76 | 450 | 35.1 | n/a | n/a | |
| C4 | D1×D2 | 0.085 - 0.6, 0.085 - 0.6 | 164 | ∞ | 10.4 | 8.1 | ∞ | 155 |
| TC3 | D3 | 0.085 | 12 | ∞ | 49.3 | 0.8 | ∞ | 101 |
| S32 | D3 | 0.25 | 12 | ∞ | 49.3 | 0.75 | ∞ | 189 |
| S33 | D3 | 0.75 | 12 | ∞ | 40.2 | 1.05 | ∞ | 1162 |
| S34 | D3 | 2.0 | 12 | ∞ | 63.8 | 2.08 | ∞ | 93077 |

**Table 1: List of SREM energy channels. The column `BIN' gives the name of the channel, the column `Logic' names the corresponding detector, the column `Discriminator Level' defines the minimum energy to be deposited, the columns `Proton Energy' and `Electron Energy' define the given energy threshold of the channel and the Single Conversion Factor (SCF) to derive differential**
15  **fluxes from count rates (from Evans et al., 2008).**

### 2.2 HEND

In addition to the SREM monitors, we have used data recorded by the High Energy Neutron Detector (HEND) [Boynton et al.,
2004] on board the Mars Odyssey spacecraft. It is composed of five separate sensors that provide measurements of neutrons
20  in the energy range from 0.4 MeV up to 15 MeV. In this study, only data from the Outer Scintillator (a veto-counter and used for anti-coincidence rejection of charged particles) in channels 9–16 is used (~195->1000 keV). This sensor is the best one for space weather studies as it is sensitive to neutrons, charged particles, and energetic photons (see more information at Sanchez-Cano et al., 2018). This instrument can be used also as a proxy for GCRs, as demonstrated in Zeitlin et al. (2010), since HEND





measure secondary particles produced by the interactions of primary energetic GCR with the spacecraft, providing indirectly

a measure of the cosmic rays (Zeitlin et al. 2010).

**2.3 Orbits**

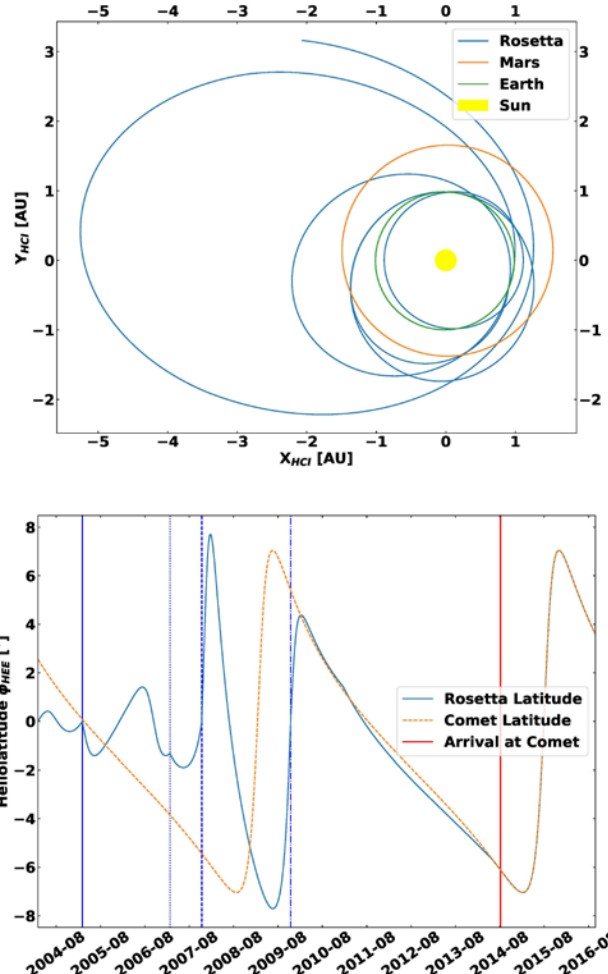

**Figure 1: Orbital information for the used data sets. Top panel: Orbits of Earth (green), Mars (orange) and Rosetta (blue) in HCI coordinates. Bottom panel: Heliolatitude in HEE coordinate system, the solid blue line indicates the first Rosetta Earth flyby, the dotted blue line indicates the Mars flyby, the dashed blue line indicates the second Earth flyby and the dashed dotted blue line**

10   **indicates the third Earth flyby.**

Figure 1 shows the orbits of Earth (green), Mars (orange) and Rosetta (blue) in Helio-Centric Inertial (HCI) coordinates. The

HCI coordinate system is defined with its x-axis pointing towards the Solar-ascending note on the ecliptic, the z-axis to be





aligned with the solar rotational axis and the y-axis completing a right-handed Cartesian triad. At scales of AU, we can assume that Earth's orbit is similar to INTEGRAL, Proba-1, Herschel and Planck's orbits and that the Mars' and HEND's orbits have also a similar orbit around the Sun. The second panel of Figure 1 illustrates the heliolatitudes travelled by the Rosetta mission, which describes how far the spacecraft and the comet travel out of the ecliptic plane. While the comet's components reflect its

periodic nature, Rosetta's components do not, since it underwent a number of orbital changes to attain the same orbit as the comet. This was achieved with several gravity assist flybys, which are indicated on the plot by vertical lines: three Earth gravity assists on 2005-03-04 (solid), 2007-11-13 (dashed) and 2009-11-13 (dashed dotted) and a Mars gravity assist on 2007-02-25 (dotted) which all had a significant impact on the trajectory of the spacecraft. The final vertical line, in red, indicates the comet rendezvous on 2014-08-06.

**2.4 Data processing**

In this section, we explain the procedure of the GCR analysis. SREM channel TC2 was chosen to be the main channel for this study, having the highest proton energy threshold of the non-coincidence counters with about 49 MeV. Since the GCR spectrum is dominated by very high energetic particles, it is therefore the most sensitive channel for these purposes. As this study focuses on a count rate spectrum consisting of GCRs, it is necessary to clean the data sets from solar proton events (SPE)

contamination, by removing intervals containing SPE events. The times were chosen based on the ESA Solar Proton Event Archive' (http://space-env.esa.int/index.php/Solar-Proton-Event-Archive.html). Since the data in this archive are based on geostationary satellites, further SPEs detected by HEND and Rosetta at locations with a significant longitudinal difference with respect to the Earth's heliocentric longitude had to be removed manually. In practice, we removed peaks associated with SPEs in data when SPEs exceeded a local daily mean value of count rates. The INTEGRAL data set needed an additional

processing to remove the signatures of Earth's inner magnetosphere trapped particle environment, by only considering spacecraft altitudes above 60,000 km from the origin of the geocentric equatorial inertial (GEI) coordinate system.

The HEND data had to be processed in multiple steps. First, the SPEs were removed in a similar procedure as for the SREM data. Second, the reconfiguration of the anti-coincidence switch on HEND in 2012 had to be taken into account. This correction

manifests itself in a constant offset of 750 counts from 2012-10-19 16:02:54 [J. J. Plaut, personal communication] which can be easily reversed. Finally, the data were converted from count to counts per second by considering a collection interval of 19.7 seconds [Zeitlin et al., 2010].

**2.5 Cross-calibration between radiation monitors**

A quantitative comparison between the measured count rates from different radiation monitors on spacecraft requires a cross-

calibration exercise. All SREM instruments were calibrated against the INTEGRAL sensor, as INTEGRAL offers the longest baseline. HEND was then calibrated to the calibrated SREM on-board Rosetta. The calibration of Rosetta to INTEGRAL was based on their hourly averaged data of three days around Rosetta's three Earth flybys (similar space radiation environment





during the flybys) on 2005-03-04, on 2007-11-13 and on 2009-11-13. A linear fit of the data sets is performed from which a fit function can be obtained. The latter is used to calibrate the Rosetta/SREM channel TC2 data. Figure 2 displays the three hourly averaged data sets with the corresponding standard error, together with the fit curve. The data appear well aligned during the three chosen calibration periods, suggesting similar response to the GCR radiation environment and good stability

5   over time between Rosetta and INTEGRAL. We associate the 2.8% difference between INTEGRAL and Rosetta, taken from the gradient fit of $1.028 \pm 0.005$, with differences in the sensitivity area of the two SREM detectors. We associate the GCR count rate changes over the years to be associated with the solar cycle (e.g. Heber and Potgieter, 2008; Potgieter, 2013) which is discussed in more depth below.

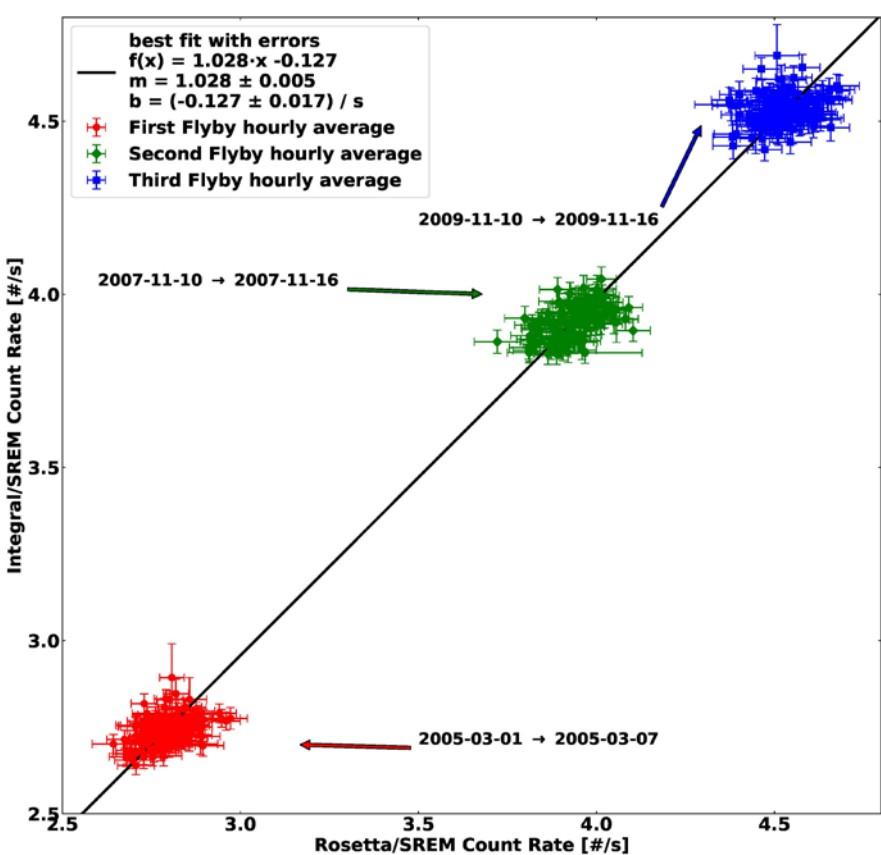

**Figure 2: Cross-calibration between INTEGRAL and Rosetta SREM instruments. Fitted data of Rosetta SREM and INTEGRAL SREM channel TC2 for the time around Rosetta's Earth flybys.**





The fit yields the calibration function of equation:

Count (Rosetta) = 1.028 x count (INTEGRAL) - 0.127 / s

This function is then applied on the whole data set of channel TC2.

*Calibration of Planck/SREM, Herschel/SREM and Proba-1/SREM to INTEGRAL/SREM*

Under the assumption that Planck, Herschel and INTEGRAL measure in a similar space radiation environment, excluding the
INTEGRAL radiation belt passages, the calibration of Planck and Herschel to INTEGRAL is based on the whole channel TC2
data set of the spacecraft at Lagrange point L2 to ensure the highest statistics and therefore most accurate fit possible. The fit
yields the following calibration functions:

Count (Herschel) = 0.931 x count (INTEGRAL) + 0.060 / s
Count (Planck) = 0.938 x count (INTEGRAL) + 0.028 / s

Cross- calibration with Proba-1 was carried out in a similar way to Planck and Herschel, although in this case, INTEGRAL
counts were consistently higher than Proba-1 by a factor of 1.256. In addition to a possible active area difference, PROBA-1's
lower count rates can easily be explained by its low altitude orbit, with the solid angle of Earth presenting a shielding for GCR
fluxes. The fraction of the solid angle divided by 4 pi is equal to 21.2 %.

The fit yields the calibration function:
Count (Proba-1) = 1.256 x count (INTEGRAL) + 0.154 / s

This function is applied on the whole data set of Proba-1's channel TC2.

*HEND*

The neutron monitor HEND is calibrated with respect to SREM-Rosetta. Assuming a mean heliocentric distance of Mars at
1.5 AU, Rosetta data were used when the spacecraft was located at the same distance from the Sun, which happened seven
times during the Rosetta cruise. These periods are indicated in Figure 3, each covering +/- 3 days around the indicated time
and made up of hourly averaged data.





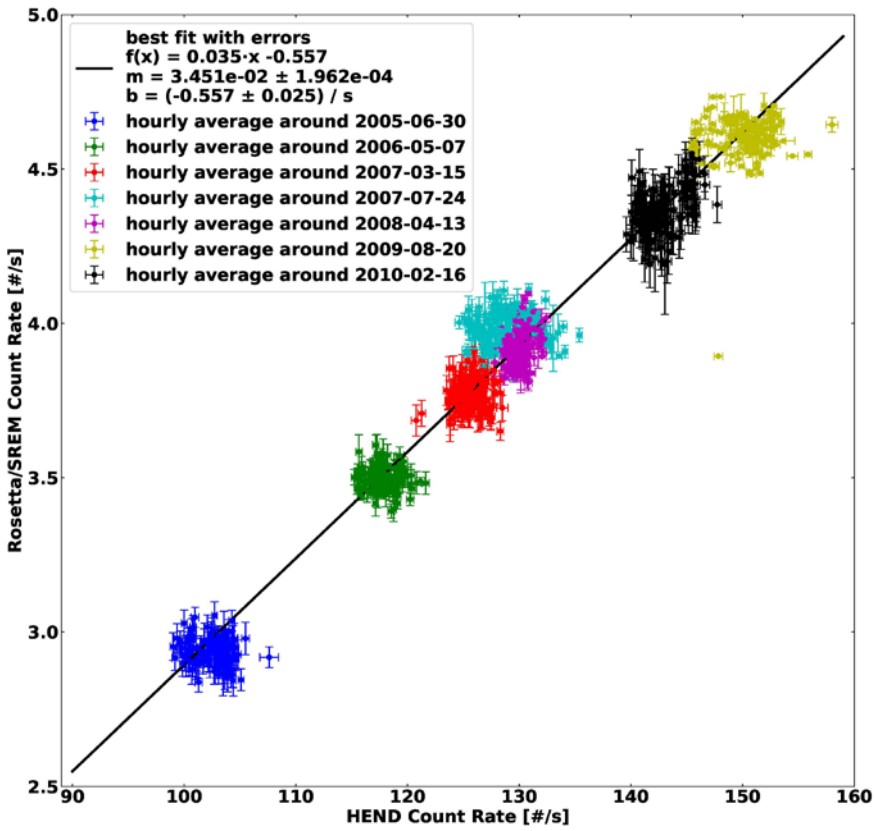

**Figure 3: Cross-calibration of Mars Odyssey HEND with Rosetta SREM. The seven groups of data correspond to the seven times Rosetta was at 1.5 AU from the Sun.**

The fit yields the calibration function:

Count (HEND) = 0.035 x count (Rosetta) - 0.557 / s

This function is applied to the whole HEND data set. It should be also noted that the shadow of Mars is not included in this study. The corresponding shielding is expected to be about 20 %.

Table 2 lists the fitting parameters, for the generic function: count (spacecraft 1) = a x count (spacecraft 2) + b

| Spacecraft 1 | Spacecraft 2 | a | b |
|---|---|---|---|
| **Rosetta** | INTEGRAL | 1.028 | -0.127 |
| **Herschel** | INTEGRAL | 0.931 | 0.060 |
| **Planck** | INTEGRAL | 0.938 | 0.028 |



| Proba-1 | INTEGRAL | 1.256 | 0.154 |
|---|---|---|---|
| **Mars Odyssey** | Rosetta | 0.035 | -0.557 |

**Table 2: Fitting parameters for the function: count (spacecraft 1) = a x count (spacecraft 2) + b**

25



# 3 Data analysis

## 3.1 Overview of the data, GCR modulation



**Figure 4: Temporal evolution of various data sets. 1st panel: SREMs and HEND count rates averaged over 27 days. 2nd panel: Rosetta**
**Heliocentric distances. 3rd panel: Interplanetary magnetic field measured by ACE at 1AU. 4th panel: Sun spot number. 5th panel:**



**Computed tilt angle of the heliospheric current sheet. The solid red vertical line indicates the minimum sunspot number while the dashed vertical line indicates the maximum sunspot number. The dash-dot green vertical line indicates the peak of the Rosetta SREM count rate. The dotted blue line indicates the reversal of polarity of the average solar polar flux.**

Having implemented the appropriate cross-calibrations, a qualitative and quantitative comparison of the obtained data sets is

possible. Data are averaged over one solar rotation (27 days) in order to minimize longitudinal effects. Such longitudinal effects are illustrated in Annex 2. In the upper panel of Figure 4, radiation data of Rosetta/SREM, INTEGRAL/SREM, Planck/SREM, Herschel/SREM, Proba-1/SREM and HEND are shown. The SREM and HEND data are very well aligned throughout the whole epoch, although some differences do stand out, in particular for HEND and Rosetta, which we associate with different heliographic locations. The other panels display the Rosetta Heliocentric distances, the interplanetary magnetic

field measured by ACE at 1AU, the Sun spot number, and the computed tilt angle of the heliospheric current sheet. The peak count rate observed at Rosetta occurs in early 2009 (vertical green line), as the spacecraft passed through the aphelion of one of its orbit around the sun. This peak occurs during the long minimum solar activity and is well correlated with the minimum of interplanetary magnetic field of ~ 4 nT. The HEND peak in late 2009 is coincident with the Rosetta peak, being about at the same heliocentric distance, and the Rosetta count rate is close to the values observed at 1 AU by INTEGRAL and Proba-1.

The relative enhancement of the Rosetta count rate in 2010 is coincident with Rosetta's outbound leg, at heliocentric distances of ~ 3.5 AU or more, shortly before rendezvous manoeuvres and hibernation and again could be associated with the radial gradient of GCRs in the inner heliosphere. However, following hibernation exit in 2014, Rosetta's SREM count rates are similar to HEND even though Rosetta is ~ 4.2 AU at this time. Shortly after, surprisingly, the values dropped below the other measurements. This behaviour is discussed in section 3.3.

The count rates from all spacecraft display a long term variation over ~ 13 years, which we compare with various solar wind parameters. The interplanetary magnetic field (IMF) and Solar wind measured by the Advanced Composition Explorer (ACE) (Stone et a., 1998; Smith et al., 1998; McComas et al., 1998) along with the tilt of the heliospheric current sheet is plotted in the other panels of Figure 4. The heliospheric current sheet (HCS) tilt is the maximum latitudinal extent of the HCS, computed

using a potential field model applied to photospheric magnetic field observations (Hoeksema, 1995; Ferreira and Potgieter, 2003), showing the known solar cycle modulation of GCRs. In addition, the expected anticorrelation between GCR and IMF and Sun spot number was calculated and the result can be found in Annex 1.

**3.2 Helioradial gradient of cosmic rays**

The availability of data from a family of instruments at different heliocentric distances allows the radial gradient of cosmic rays to be examined, providing an insight into the behaviour of the galactic cosmic ray propagation between 1 and 4.5 AU. The cosmic ray radial gradient is computed following the equation (Webber and Lockwood, 1991):

$$Gr = \ln (N2/N1) / (r2-r1) \qquad (1)$$



Where N is the count rate and r is the heliocentric radial distance at locations 1 and 2, where r2 > r1.

The radial gradient was computed from the INTEGRAL and Rosetta data set for selected periods of the Rosetta mission (e.g.

in between planetary flybys), and the results are summarised in Table 3, which contains also some key heliophysic parameters.

| Period | Rosetta heliocentric distance [AU] | Solar activity | Range of IMF at 1 AU [nT] | Range of tilt angle [degrees] | Radial gradient [%/AU] |
|---|---|---|---|---|---|
| 2005-07-01 to 2006-06-30 | 1.43-1.75 | Low | 4.39-6.70 | 9.70-24.10 | 1.68±0.44 |
| 2007-01-01 to 2007-10-31 | 1.08-1.59 | Low | 3.91-5.14 | 11.30-15.90 | 2.59±0.48 |
| 2008-03-01 to 2009-10-31 | 1.10-2.26 | Minimum | 3.56-4.34 | 4.50-17.60 | 3.16±0.16 |
| 2010-01-01 to 2011-06-30 | 1.13-4.43 | Medium | 3.95-6.27 | 17.80-64.10 | 3.16±0.17 |
| 2014-01-01 to 2014-03-17 | 4.31-4.41 | High | 4.68-6.98 | 54.40-70.50 | 2.13±0.09 |

**Table 3: Radial gradients obtained for a given Rosetta-Sun distance, solar activity, interplanetary magnetic field and computed tilt angle for the mentioned periods. There are no obvious correlations between the radial gradient and the heliophysics parameters.**

Using equation (1) we consider the evolution of the radial gradient between Rosetta and INTEGRAL for the entire mission in

Figure 5, where the different coloured points indicate the different phase of the mission. Blue are pre-hibernation data, orange

are January-July 2014 and green are July 2014-September 2016 data. A fit has been computed to the pre-hibernation data (red

line) and the July 2014-September 2016 data (black line). During the pre-hibernation phase, the slope, which corresponds to

the radial gradient, is found to be 2.96±0.12 %/AU. This result agrees well with previous studies (e.g. Vos and Potgieter, 2016;

Gieseler and Heber, 2016). The slope during the comet phase was found to be -2.8±0.12 %/AU. In Figure 6, the count rate

variation at Rosetta and INTEGRAL are shown. The drop in the count rate occurs during the approach phase, between February

and May 2014. After that period, the count rate variation and the ratio is in very good agreement with the expectation of a

positive radial gradient of about 2.9 %/AU (e.g. Vos and Potgieter, 2016; Gieseler and Heber, 2016).





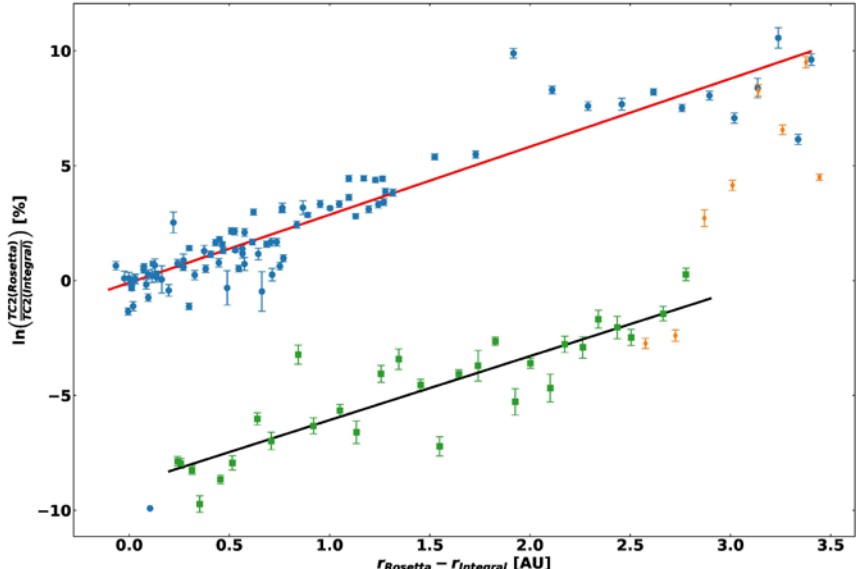

**Figure 5: Logarithmic ratio of Rosetta and Integral SREM TC2 data drawn against the difference in heliocentric distance of Rosetta and Integral. The data in blue indicates the time before Rosetta's hibernation mode, the data in orange indicates the time right after hibernation mode until end of July 2014 and the data in green are from August 2014 until the end of the Rosetta mission in September 2016. The performed fits in red and black yield the corresponding radial gradients.**

## 3.3 Apparent attenuation of galactic cosmic ray flux in the vicinity of 67P

This section discusses the relative change in GCR counts at Rosetta compared to INTEGRAL during the comet phase of the mission in 2014. This change of behavior can be observed on Figure 6. The Rosetta counts, initially above INTEGRAL, rapidly decrease and remain below INTEGRAL for the rest of the time period. This change is illustrated in Figure 5 by the dark and red fits. A similar behaviour can be observed in all three channels/detectors of SREM. Comparing the two fits (red and dark lines), the GCR fluxes after August 2014 are ~8 % lower than expected from the pre- July 2014 data.





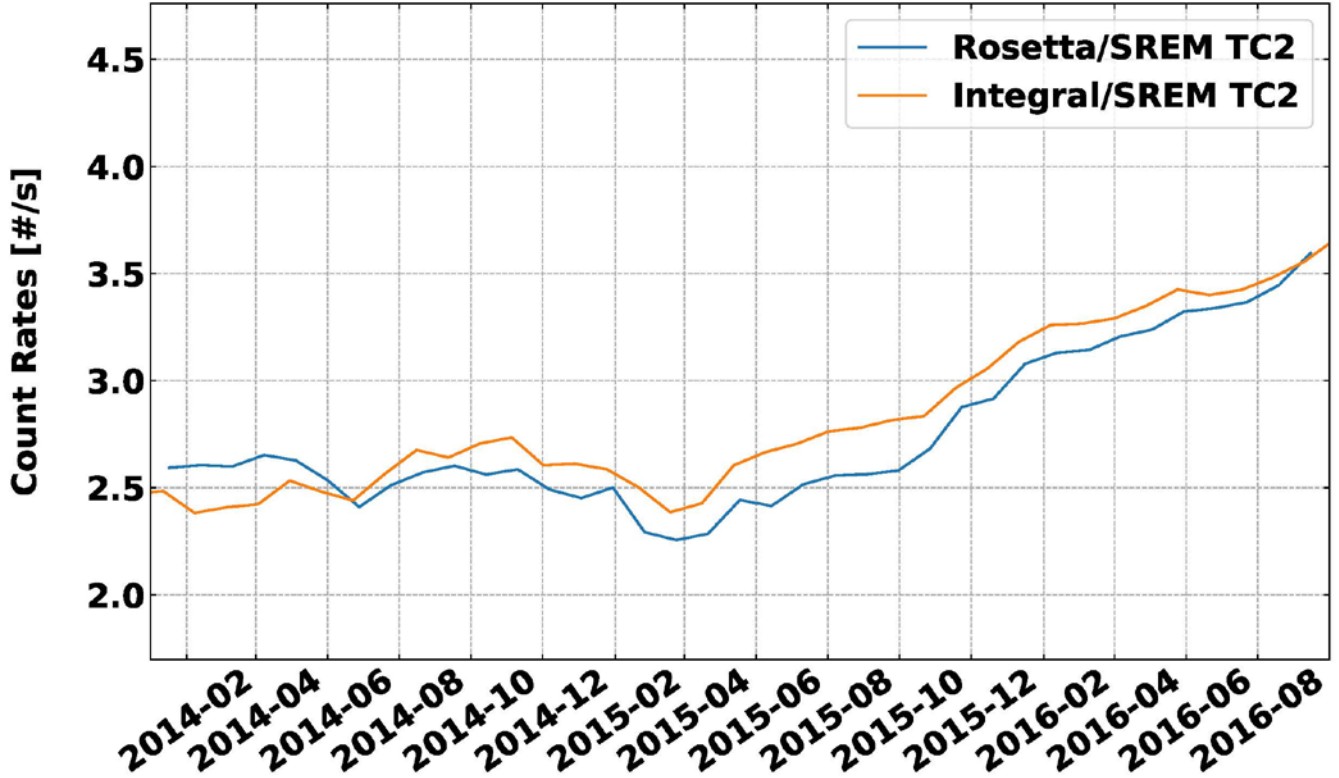

**Figure 6: Zoom on the INTEGRAL and Rosetta SREM count rates during the period of the nominal Rosetta mission. The Rosetta data clearly goes below INTEGRAL in spring 2014.**

5    In order to discuss different reasons for this apparent attenuation, we looked for changes in environmental conditions. The attenuation effect coincides with the overall Solar polarity change (the transition from a A<0 to a A>0 cycle). Previous studies have indicated a dependence of GCR fluxes with Solar polarity (e.g. Potgieter, Burger and Ferreira, 2001) with radial gradients being smaller during A>0 cycles. Negative latitudinal gradients have been reported (e.g. Potgieter, Burger and Ferreira, 2001), but only a fraction of one % per degree (Gieseler and Heber 2016). During the comet phase, Rosetta moved from around -7.5

10   ° to +7.5 ° heliolatitude, which could not account for the decrease in GCR fluxes. However, latitudinal gradients have only been reported during A<0 cycles, as opposed to the cycle 24, where A>0.

The decreasing ratio begins when Rosetta reaches around 20,000 km from the cometary nucleus and persists more or less at the same level until the end of the Rosetta mission. We cannot discern any anomalous Rosetta SREM instrumentation

15   behaviour during the comet phase. For example, the period May-July 2014 coincided with several large rendezvous manoeuvres, where hundreds of kg of propellant material were used. A similar (in magnitude) series of manoeuvres were also implemented prior to hibernation in early 2011 suggesting thruster induced contamination or deterioration of the SREM



detectors is not responsible. In addition, one would not expect the reduction of propellant within the fuel tanks to increase shielding. We note that INTEGRAL count rates are also consistent with Proba-1 measurements during this period, suggesting both instruments are behaving nominally.

We have considered the solid angle presented by the nucleus to have some impact on counts, with the comet angular size getting as high as 30° in November 2014 during lander delivery and ~ 70° in September 2016. However, a majority of the time the angular size was < 10 °, and insignificant (<< 10°) when the "attenuation" began in early 2014, suggesting the nucleus is not a major driver here.

Ground based measurements of the comet indicate cometary activity already began in February 2014 (Snodgrass et al., 2016) with Rosetta remote observations by the OSIRIS camera being able to resolve coma activity in March-April 2014, indicating a coma extent of around 1000 km at that time (Tubiana et al., 2015). However, it was not until August that the in-situ instruments onboard Rosetta began to discern a coma signal, when the spacecraft got to within 100 km of the nucleus (Altwegg et al., 2015; Rotundi et al., 2015) so, the transition in behaviour occurs before the spacecraft is immersed in the cometary coma.

Nucleus activity and coma extent increases significantly in the subsequent months (e.g Hansen et al., 2016) yet with no corresponding change in the gradient of GCR over this time. However, the potential shielding of the cometary gas and dust and associated plasma environment cannot be fully ruled out.

## 4 Discussion and concluding remarks

In this study, we have analysed data from the SREM instruments onboard several ESA spacecraft as well as the HEND

instrument onboard Mars Odyssey. The combination of all these different instruments give us multi-point observations of GCR within the Solar System, which constitute a very useful and rich dataset. It is important to note that the primary purpose of this dataset is engineering. However, they highly valuable for pure scientific studies as illustrated in this paper. Our first step was to calibrate the different SREM sensors onboard different spacecraft, such as ROSETTA, INTEGRAL, HERCHEL, PLANCK, and PROBA-1. Then, the ROSETTA data was also calibrated with respect to HEND on board Mars Odyssey at Mars' distance.

In addition, the data are averaged over a solar rotation period of 27 days, in order to avoid longitudinal effects. However, not doing so allows to study time shifts between solar wind features between Earth and another location, as illustrated in annex 2. As a result, we have obtained a very useful dataset, totally calibrated, that give us information of the evolution of GCR with the solar cycle and heliocentric distance evolution. Some additional information regarding the GCR variability with respect to the interplanetary magnetic field (IMF) and sun spot number (SSN) can be found in the annex.


We have also demonstrate the value of the combination of such data sets in giving a broad view of the distribution of galactic cosmic rays in the inner heliosphere, both geographically and temporally. An important point has been the confirmation of the



modulation of galactic cosmic rays with respect to solar activity, as well as the anticorrelation with the interplanetary magnetic field. Also, thanks to the unique Rosetta trajectory within the inner Solar System, the helioradial gradient of galactic cosmic rays between 1 and 4.5 AU was found to be 2.96 %/AU, matching previous reports (e.g. Vos and Potgeiter, 2016). This information provide insights into the behaviour of the galactic cosmic ray propagation within the inner heliosphere.

When considering the cometary phase of the Rosetta mission, from early 2014 to September 2016, the radial gradient changed, equivalent to an overall 8% attenuation in count rate, and reversed, with count rates at INTEGRAL persistently greater than those at Rosetta, contrary to general expectations. We have considered several potential influences on these measurements to explain this observation, including heliospheric and more local environment conditions. Although several aspects can be

discounted for the GCR reduction in the comet environment, further work needs to be carried out on the nature of the overall cometary coma characteristics to quantify its potential impact, along with heliospheric GCR modulation associated with the solar polarity changes. The combination of the extended minimum of Solar cycle 23 with the weakest Solar maximum (cycle 24) for a century, coincident with the time period under scrutiny will also be examined.

In addition, other possible follow-up studies include a detailed temporal and spatial analysis of all the radiation datasets, as well as short scale variations of the GCR flux between close points, such as between Earth and Lagrange point L2, or when Rosetta did a flyby to Earth and Mars.

**Acknowledgements**

T. Honig acknowledges the ESA stagiaire program. The authors thank Oldenburg colleagues for useful discussions. The SREM data are available at https://spitfire.estec.esa.int/ODI/dplot_SREM.html. The HEND data are available at NASA PDS. The ESA Solar proton event archive can be found at: http://space-env.esa.int/index.php/Solar-Proton-Event-Archive.html. HCS and Solar polar field data are available at http://wso.stanford.edu. Rosetta is an ESA mission with contributions from its member states and NASA. B.S.-C., acknowledges support through STFC grant ST/S000429/1 .




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

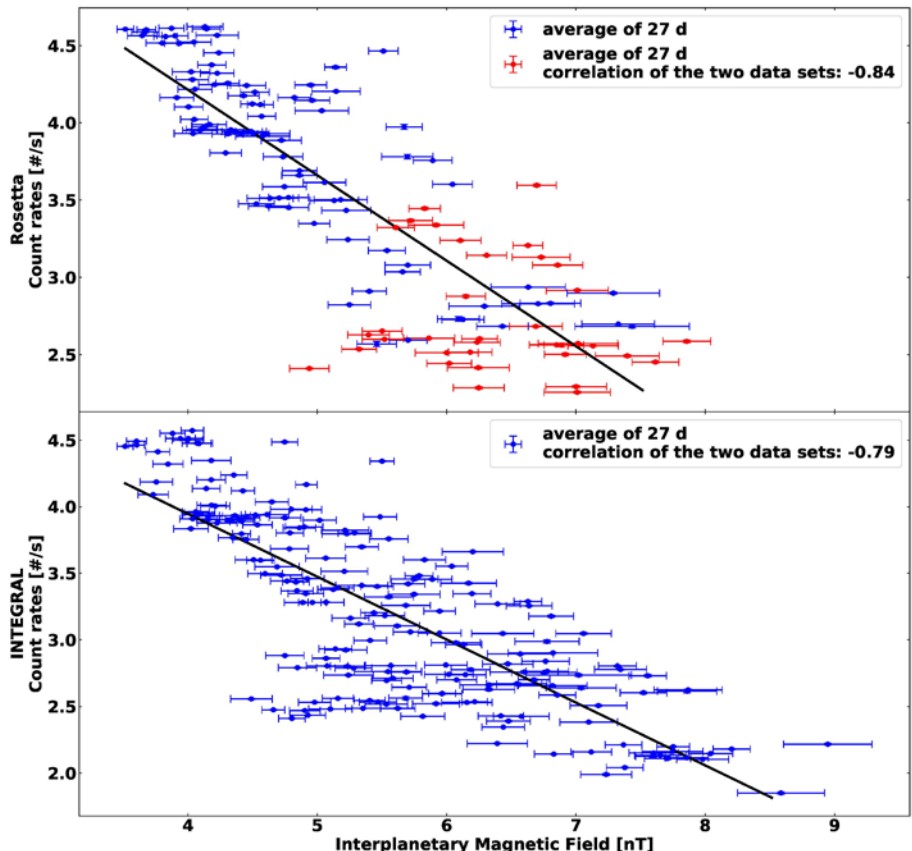

10    **Figure Annex 1: Anticorrelation of Rosetta and INTEGRAL SREM data with the IMF. The error bars for all data points correspond to the standard deviation. The Rosetta SREM data in the first panel is distinguished into pre-hibernation phase (blue) and post-hibernation phase (red).**

| Data set | Rosetta | INTEGRAL | Planck | Herschel | HEND | Proba-1 |
| --- | --- | --- | --- | --- | --- | --- |



| Period | 2004-10-21 to 2016-09-15 | 2002-10-17 to 20017-02-18 | 2009-05-14 to 2013-09-23 | 2009-05-14 to 2013-06-07 | 2002-01-14 to 2016-06-14 | 2001-12-10 to 2017-03-30 |
|---|---|---|---|---|---|---|
| **IMF** | -0.84 | -0.79 | -0.67 | -0.73 | -0.75 | -0.78 |
| **SSN** | -0.78 | -0.67 | -0.81 | -0.81 | -0.77 | -0.60 |

**Table Annex -1: Correlation coefficients calculated based on 27 day averaged data from the radiation monitors.**

The correlation coefficients, listed in Table Annex-1, show the expected anticorrelation (e.g Cane et al., 1999; Belov et al., 2000 and Utomo, 2017). IMF comparisons have a stronger correlation than the Sun spot number at Rosetta, Integral and Proba-

5   1 than Planck, Herschel and HEND. Planck and Herschel comparisons are over a shorter time scale during the rising phase of solar cycle 24, and HEND comparisons may be complicated by its indirect measurements of GCRs. Overall, however, the expected trends are well present.



**Annex 2: Time shift of solar wind features**

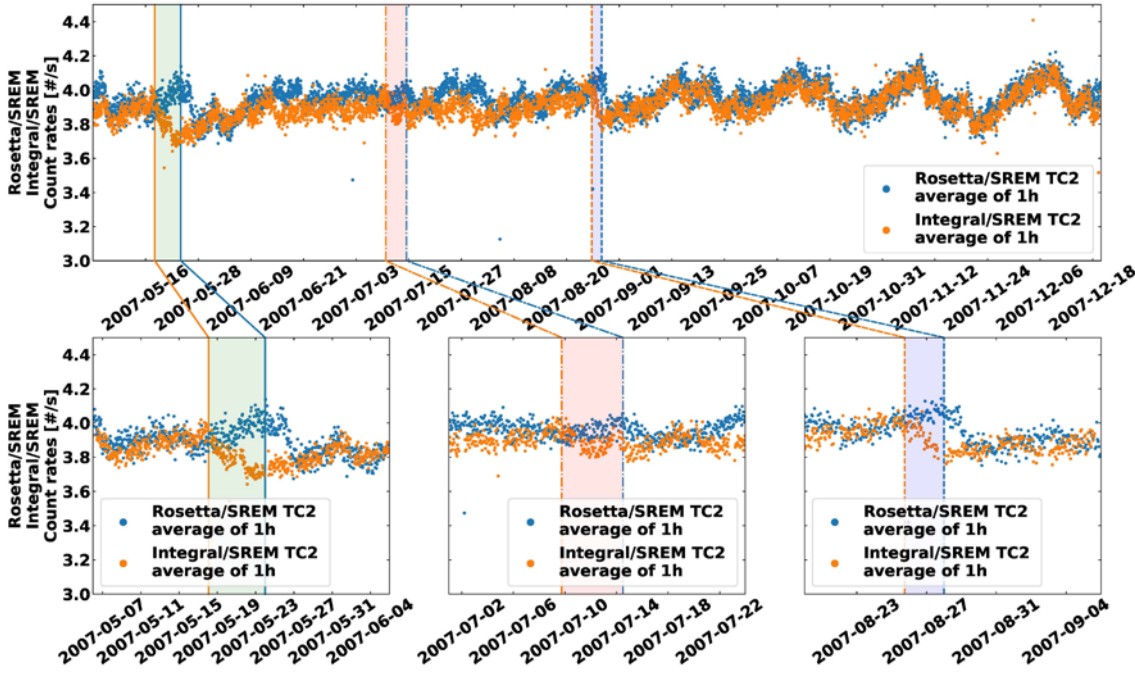

**Figure Annex 2: Solar wind feature shifts. First panel: count rates of Rosetta SREM TC2 (blue) and INTEGRAL SREM**

**TC2 (orange) from May until December 2007. Second panel: Zoom in distributions of the three periods marked with vertical lines.**

GCR short temporal variations can be driven by coronal mass ejections (CMEs) and corotating interaction regions (CIRs) [e.g. Moraal, 2013; Badrudin and Kumar, 2016; Sanchez-Cano et al., 2017; Witasse et al., 2017] and can influence the timing of

signals at various locations in the heliosphere. To demonstrate this, we examine Rosetta and INTEGRAL data during the period from mid until end of 2007. In the first panel of Figure Annex-2, the count rates of channel TC2 of Rosetta (blue) and INTEGRAL (orange) are shown. The temporal delay in the measurement from the two spacecraft is clearly visible and decrease with time. The other panels display a zoomed window of three periods, where correlated features or peaks are indicated in the corresponding data sets by straight, point dashed and dashed vertical lines with INTEGRAL in orange and Rosetta in blue. In

May 2007, Rosetta was around 1.58 AU from the Sun and separated in longitude form the Earth by about 60°. In July 2007, Rosetta was around 1.55 AU from the Sun and longitudinally ~45° from Earth. Finally, in August 2007, Rosetta was around 1.4 AU and only ~15° from Earth longitudinally. For the first event, the delay between INTEGRAL and Rosetta is six days and two hours, for the second event, four days and 18 hours and the third event, in August 2007, two days and six hours. These



variations are related to the changing relative location/longitude of the spacecraft and Parker spiral configuration. In order to avoid these longitudinal effects, the data are averaged over 27 days (see section 3.1).