# Peer review of "Multi-point galactic cosmic rays measurements between 1 and 4.5 AU over a full Solar cycle"

_Annales Geophysicae, 2019_

## Short Comment (SC1) · 14 May 2019

This is a quite interesting study which I enjoyed reading. Some brief comments are below:

1) Equation 1 is used to estimate radial gradients. However, N1 & N2 are count-rates, which are proportional to integral fluxes. Therefore, the estimated parameter is an "integral gradient". "Differential gradients" require to have differential flux measurements. For instance, it is my understanding that Gieseler & Heber (2016) estimate differential gradients, so comparison with the values obtained in this study should be reconsidered, even if values are similar.

2) Both differential and integral gradients have an energy dependence. For the latter,

which are more relevant to the present study, it matters above which energy fluxes are integrated. The used channel captures protons >49 MeV, however, from other SREM papers it seems that the geometry factor <100 MeV is rather low. So, I assume the estimated gradients have are for protons much above 100 MeV. Maybe folding the response function of the TS2 channel with a standard GCR spectrum can show which energies dominate.

3) I am not sure how the HEND data are used in the study. In order for them to be compared with those from SREM, they have to be normalized to the INTEGRAL count-rates, since SREM data are normalized to the INTEGRAL measurements. This means that in the y-axis of Fig. 3, one should used the INTEGRAL-normalized rates of SREM, not the raw SREM rates. I.e. this has to be a 2-step normalization. If that was actually done, its has to be clarified in the text.

4) After HEND data are normalized to SREM, they were not used in any part of the analysis. E.g. they may also be used to estimate radial gradients, which should be similar to those coming from the SREM/INTEGRAL ratios, otherwise they may be indicative of uncertainties in the gradient estimation, or, even better, of a radial dependence of the ratios. Instead, HEND are only mentioned briefly in lines 5-15 in p.11.

5) In addition to the comment above, it is clear that in the comet phase, where SREM sees a negative radial GCR gradient, the gradient between INTEGRAL/HEND is clearly positive, even if normalization may require an update (see comment 3). That further supports the possibility of a reduction of GCR fluxes around the comet. My suggestion is the following: a)Estimate the radial gradient between INTEGRAL/HEND for times during Rosetta's comet phase b)From this radial gradient, estimate what should have been the count rate of SREM c)Estimate the difference between the expected and the measured count-rate d)This difference may be estimated also by using in step (b) the average positive radial gradient as found from the data shown in Fig. 5 e) Then, the difference (estimated by any of the methods) could be organized as a function of heliocentric distance (essentially activity) or any other relevant parameter. It appears

intriguing that in Fig. 6, the count-rate difference appears to maximize around mid-2015, close to perihelion, and tends to become zero again towards the end of the mission.

I hope the authors find these comments helpful.

---

## Short Comment (SC2) · 20 May 2019

Overall, the paper presents interesting findings. However, a more detailed description of the physical processes leading to eg the gradient in GCR flux would improve the scientific content significantly.

Figure 1a: the green and blue are hard to distinguish, maybe another color combination would be better here

1. It is unclear to me why the instruments need cross-calibration. What are the technical reasons for the instruments different behaviour if they are essentially the same model? You mention sensitivity area, can you elaborate further? Do you have any reason to believe that the dependence of the countrate of two instruments is linear?

[Figure]

Could it also be something else? (second or third order?)

2. p7l3: the equation given here is not consistent with what is shown in Figure 2. From figure two the relationship should actually be: Count(Integral)=1.028 x count(Rosetta) -0.127 . Then all the other calibration functions should also be checked.

3. p11l21ff: The correlation is very obvious. You say this is something that was expected and address this briefly in the annex. I think it would be better suited here and needs to be explained in more scientific detail.

4. p12l11ff: Again, a physical explanation of why this gradient is expected would be good.

5. p15l22: However, they ARE highly.....

6. p15l31: demonstrate –> demonstrated

---

## Referee Comment (RC1) · Anonymous Referee #1 · 4 Jun 2019

General comments

This paper presents a study of the different high energy particle deterctors which have been mounted on severl different ESA missions. As these missions have different objectives multi point measurements of Galactic Cosmic Rays are possible in the heliosphere. The paper is clear and well written and should be published in Annales Geophysicae after the following minor comments have been addressed

Minor comments

typo and spacecraft's component material->and a spacecraft's component materials

Figure 1b the lines are a little difficult to distinguish on the printed page and the dot dash and dashed lines are difficult to distinguish here.

[Figure]

p5l19 I would like more details on the process to remove SPEs. Anything above a local average is removed? Would a method such as a hampel filter be more appropriate here? I would like a little more detail here.

Figures 2 and 3 can you quote a value for the goodness of fit like chiˆ2

Figure 4 and the related discussion the sun spot number is displayed but there is no source for this data. There are several different metrics which can be used as a 'sunspot number' see Lockwood 2014 and refs therein https://doi.org/10.1002/2014JA019970

Figure 4 what is the cadence of the data in Figure 4, are these averaged with over 27 days also? It would be interesting to also plot the variance or the stadard deviation for the same window width as the averaging of the magnetic field as a proxy for the fluctuation amplitude of the magnetic field fluctuations.

---

## Referee Comment (RC2) · Anonymous Referee #2 · 29 Jun 2019

**SUMMARY**

This manuscript is based on galactic cosmic ray measurements provided by the HEND instrument onboard Mars Odyssey and the SREM radiation monitors onboard Rosetta, and four different near-Earth s/c (INTEGRAL, Herschel, Planck and Proba-1). The period analyzed (2004-2016) covers more than one solar cycle and includes the Rosetta encounter with comet 67P/Churyumov-Gerasimenko. After removal of Solar energetic particle events, a cross-calibration of the counting rates provided by the different instruments is performed. SREM onboard near-Earth s/c are cross calibrated using INTEGRAL as reference measurement. Rosetta flybys of the Earth are used to cross-calibrate Rosetta/SREM and INTEGRAL/SREM. Finally, cross-calibration between Rosetta/SREM and Mars Odyssey/HEND is achieved using Rosetta data at heliocentric distances near 1.5 AU. As expected, cosmic ray fluxes at all locations anti-correlate with markers of the solar activity cycle (sunspot number and interplanetary magnetic field). Differences in the GCR rates measured near the Earth, at Mars and by Rosetta before hibernation are interpreted in terms of positive GCR radial gradients, with values consistent with previous studies. After hibernation, around the encounter of Rosetta and comet 67/P, an unexpected reduction of the counting rates at Rosetta compared to near-Earth measurements is reported. The authors briefly enumerate/discuss possible origins such as a negative GCR latitudinal gradient (unlikely, due to the small helio-latitudinal interval swept by the s/c), effects related to mission operations (mostly discarded by the authors) or attenuation by the plasma environment around the cometary coma.

Although the authors are rather inconclusive about the causes of the apparent decrease in the GCR flux around the comet encounter and leave this question open for future studies, the manuscript presents relevant new data. This study illustrates the scientific potential of the multiple SREM datasets, primarily intended for engineering purposes, providing cross-calibration factors valuable for future multi-point studies. GCR radial gradients constitute a relevant input for GCR modulation investigations.

I found this work interesting and adequate for publication in Annales Geophysicae once the following comments have been addressed in a revised version of the manuscript.

GENERAL COMMENTS

SREM TC2 channel is a single detector channel. Since no coincidence/veto logics are applied, this channel could include a significant contribution from secondary particles induced by cosmic ray interaction with the s/c. Authors should at least briefly discuss if this is the case, as well as the relative importance of possible sources of background. Since the study covers a long time interval and a relatively wide range of radial distances, spatial and/or temporal variations of the background can be a critical issue for the analysis of quiet-time fluxes (GCR) and could affect the reliability of

cross-calibrations along the period under analysis.

SPECIFIC COMMENTS

Page 3, line 5. Although single detector channels such as TC2 are omnidirectional, it would be helpful for the later interpretation of the data to briefly describe the pointing of the Rosetta SREM aperture during the cruise phase and during the comet encounter.

P3 Table 1. The second column (logic) lists "D1" as logic for channels TC2 and S25. This seems inconsistent with e.g. page 2 line 26 and with Table 1 in Evans et al., 2008. Please check/correct since channel TC2 is the basis of the study presented in the manuscript. Instead of just providing a nominal "49 MeV to infinity" energy range, it would be valuable to discuss the rigidity or energy range roughly represented by TC2 counting rates (taking into account the typical shape of the modulated GCR spectrum above 49 MeV and probably the energy-dependent instrument response). This information would be quite useful when comparing the inferred radial gradient with previously published results (Page 12).

P4 Figure 1, bottom panel. Please define/clarify HEE, since this acronym normally stands for the Heliocentric Earth Ecliptic coordinate system and therefore latitudes would correspond to ecliptic latitudes and not to heliographic latitudes. Do the authors mean HEEQ (Heliocentric Earth Equatorial)? If not, replace "heliolatitude" by "ecliptic latitude". This difference is also relevant for the discussion in P14L10.

P5L19. The SREM channels and the exact procedure used to manually filter solar energetic particle events should be specified. Some events could be difficult to detect in TC2 (but still contribute) but become clearly visible at energies below 49 MeV.

P5L25. The reconfiguration of HEND should be explained with further detail. What is the ultimate reason for the count rate increase/offset?

P6L5-6. "We associate the 2.8% difference……with differences in the sensitivity area of the two SREM detectors". Other factors such as noise levels, obstructions or different

s/c mass distribution around the sensor head could contribute to this small difference between the nominally identical units onboard INTEGRAL and Rosetta (as well as Herschel, Planck and Proba-1).

P7L23. Is this equation correct? The preceding text mentions that Proba-1 counting rates are systematically lower than INTEGRAL counting rates, then why is INTEGRAL rate (and not Proba-1 rate) multiplied by a number >1 in order to obtain a "corrected" Proba-1 rate directly comparable with INTEGRAL? The same question applies to the rest of cross-calibration equations. Probably this is just a notation problem, but it should be checked.

P8L6. See comment P7L23 above and clarify how the calibration factors are applied to HEND data in order to make them directly comparable with Rosetta.

P10 Figure 4. What is the reason for the peaks observed in HEND rates shortly after the vertical red dashed line? The difference between the near-Mars HEND and near-Earth SREM rates seem larger at the end of the plot (2015-16) than at the beginning (2003), while solar activity levels are comparable (although with opposite solar polarity). This, together with the comment on P5L25 raises some doubts about the stability of cross-calibration. Some discussion of these differences could be introduced e.g. in P11L7-8. Visibility of the different lines in the first panel could be improved.

P12L13-14. When citing agreement with Vos and Potgieter, 2016 and Gieseler and Heber, 2016, please mention the rigidity/energy ranges studied by these authors. The radial gradients presented here are based on counting rates accumulated over a broad energy range, which makes difficult to define a reference energy for comparison (see comment about Table 1 above).

P12L16-17 and P14 Figure 6. "After that period, the count rate variation is in very good agreement with the expectation of a positive(?) radial gradient...". This sentence sounds quite confusing since the Rosetta rates remain below the INTEGRAL rates till late 2016 (Figures 4 and 5 and discussion in following pages). In order to make

easier the interpretation of Figure 6, I strongly suggest including panels showing the heliocentric distance of Rosetta and the distance between Rosetta and Comet 64P. In order to keep consistency with Figure 4, I also suggest plotting INTEGRAL SREM rates in green color.

P14L13. "The decreasing ratio begins when Rosetta reaches around 20,000 km from the cometary nucleus". See comment above about including distance between Rosetta and the comet in Figure 6.

P14 and P15. Please briefly mention that the counting rates at Mars/HEND always stay higher than those registered near the Earth (Figure 4), even during the period shown in Figure 6. This is consistent with a permanent positive GCR radial gradient and supports that the reduction in the GCR rates at Rosetta compared to Earth is related to the comet approach.

P15L16-17. In order to substantiate this suggestion, the authors should consider including and discussing the local plasma and magnetic field observations by Rosetta during the period shown in Figure 6.

MINOR AND TYPOGRAPHIC COMMENTS

Affiliations: Please replace "Universitycity" by "University".

Page 1, Line 21. "Major sources of this radiation are the Van Allen radiation belts,...". The preceding sentences put the focus on missions outside the Earth's magnetosphere, therefore this reference to radiation belts seems out of place.

P2L4. "cover a range of heliocentric distances up to 3.5 AU". Since Rosetta reached 4.5 AU, the term "heliocentric distance" sounds confusing here. Do the authors refer to the difference between the radial distances of two s/c, rather than to the Sun-s/c distance?

P2L9. "with two of them still operating...". Please specify which ones.

[Figure]

P2L10. I suggest replacing "high energetic charged particles" by "high-energy charged particles" here and elsewhere in the manuscript.

P3 Table 1. I suggest avoiding the redundant "MeV" labelling in the first and second line of the header, e.g. keeping it only in the first line.

P3L21. "This sensor is the best one for space weather...". Please reformulate this sentence in a more specific way.

P4L11. Replace "Figure 1 shows...." by "Top panel of Figure 1 shows...".

P5L5. Please, replace "and HEND's orbits..." by "and Mars Odyssey's orbits...".

P5L15. What is the energy and/or intensity threshold of the Solar Proton Event list used here?

P6L3. I suggest replacing "the fit curve" by "the linear fit".

P7L20. Since the energy threshold of SREM TC2 is relatively low (49 MeV), magneto-spheric shielding of the lower energy part of the GCR spectrum could also play a role here.

P7L39. I suggest replacing "The neutron monitor HEND" by "The HEND neutron detector" or just "HEND- Mars Odyssey".

P8L8. Indeed, the possible effect of Mars shadow would be implicitly included in the empirical cross-calibration procedure.

P8 Table 2. Please add the units of b coefficient in the first row and include a and b uncertainties in the table.

P10 Figure 4 caption, P11L10 and P11L27. Replace "sun spot" by "sunspot".

P11L5 and P15L25 "solar rotation (27 days)" While this is OK for the purpose of smoothing longitudinal and transient variations at all locations, 27 days is just the Carrington (synodic and at intermediate latitude) solar rotation period. Therefore, I suggest

removing "one solar rotation" and leaving just "27 days".

P11L9 "different heliographic locations". Do the authors mean "heliospheric locations"?

P11L12 "one of its orbits".

P11L20 "long-term".

P11L27 "anticorrelation...was calculated". I suggest replacing either by "anticorrelation...was analyzed" or by "correlation coefficient....was calculated".

P12L14 and P13L10 the term "comet phase" should be clarified/defined.

P13L12. Replace "dark" by "black".

P14L16 "hundreds of kg of propellant...etc.". Could this mass loss significantly change the mass environment around the SREM detector and reduce the rate of secondaries contributing to TC2? For completeness, the authors could also mention the separation date of the Philae module and its (small) mass.

P15L22 "they are highly...".

P15L29 "sunspot".

P15L29. Please replace "annex" by "Annex 1".

P15L30 Please replace "geographically" by "spatially".

P15L31 "demonstrated".

P16L3. The period corresponding to this radial gradient should be mentioned here.

P16L4 "this information provides".

P18L27 Replace "cosmis" by "cosmic".

P19L24 Replace "intergral" by "INTEGRAL".

P22 Figure Annex 2. A panel showing the azimuthal separation between Rosetta and

the Earth could be added to illustrate the origin of the time delays in the observed co-rotating structures.

---

## Author Comment (AC1) · 1 Jul 2019

**Overall, the paper presents interesting findings. However, a more detailed description of the physical processes leading to eg the gradient in GCR flux would improve the scientific content significantly.**

*We thank Charlotte Goetz for her feedback.*

*We plan to update the introduction in the revised article. The following the sentence "The variation of GCRs as a function of different factors (solar cycle, heliocentric distance, solar wind conditions) is an interesting topic to explore, and lead to a better understanding of the heliosphere" is replaced by:*

*"The variation in galactic cosmic rays intensity depends on different physical processes: inward diffusion in the interplanetary magnetic field, adiabatic cooling, outward convection and deceleration in the solar wind plasma, drift along the heliospheric current sheet, and interaction with magnetic structures in shocks and in interplanetary coronal mass ejections (e.g. McKibben; Potgieter, 2013; Morral 2013; Alania et al., 2014; Kozai et al. 2014; Giseler and Heber 2016). The GCR intensity is therefore varying with the solar wind velocity, the magnitude of the interplanetary magnetic field, solar activity, the heliospheric current sheet tilt angle, and the solar polarity change."*

**Figure 1a: the green and blue are hard to distinguish, maybe another color combination would be better here.**

*The figure was updated. See below.*

[Figure]

**1. It is unclear to me why the instruments need cross-calibration. What are the technical**

**reasons for the instruments different behaviour if they are essentially the same model? You mention sensitivity area, can you elaborate further? Do you have any reason to believe that the dependence of the countrate of two instruments is linear? Could it also be something else? (second or third order?)**

*Even if the radiation monitors are of the same design, they are not identical copies. It is therefore reasonable to assume that they will have similar performances, but not equal. In fact, the purpose of the cross-calibration is twofold:*

   a) *Show that the radiation monitor family behave similarly. This is a good engineering achievement, useful to report.*
   b) *Normalise the count rate in case quantitative cross-mission studies are performed, like in the present article.*

*In the article, we elaborate further. Regarding the linearity of the calibration. Second or higher orders are always possible, but looking at the behaviour (like see the plots below for Herschel, Planck, INTEGRAL and Proba-1 monitors), a linear fit appears to us very reasonable to consider.*

[Figure]

[Figure]

**2. p7l3: the equation given here is not consistent with what is shown in Figure 2. From figure two the relationship should actually be: Count(Integral)=1.028 x count(Rosetta) -0.127 . Then all the other calibration functions should also be checked.**

*The legend is misleading, but the relationship is correct. Integral = 1.028 Rosetta – 0.127, x being the horizontal axis. We will correct the legend with Rosetta and Integral replacing x and f(x).*

**3. p11l21ff: The correlation is very obvious. You say this is something that was expected and address this briefly in the annex. I think it would be better suited here and needs to be explained in more scientific detail.**

*Page 12, we add the following sentence: "This anticorrelation is due to the modulation of GCR intensity. The GCR intensity decreases when the magnetic field and the solar activity increase due to the GCR diffusion in the solar wind. This "engineering" data is a new data set that can be useful to study the this modulation.", after "In addition, the expected anticorrelation between GCR and IMF and Sun spot number was calculated and the result can be found in Annex 1."*

**4. p12l11ff: Again, a physical explanation of why this gradient is expected would be good.**

*This will be added. It is here only a question of heliocentric distance. GCRs propagate in the solar system, and their flux decrease with the solar distance, due to the increasing strength of the magnetic field.*

**5. p15l22: However, they ARE highly.....**
**6. p15l31: demonstrate –> demonstrated**

*This will be corrected in the updated version of the manuscript.*

---

## Author Comment (AC2) · 7 Aug 2019

**Replies to referee #1**

*We thank the referee for the careful review and all these very helpful comments. Our answers are in italic blue colour.*

**Typo: and spacecraft's component material->and a spacecraft's component materials**

*Corrected.*

**Figure 1b the lines are a little difficult to distinguish on the printed page and the dot dash and dashed lines are difficult to distinguish here.**

*Figure 1b has been updated as follows:*

[Figure]

*Figure 1: New Figure 1b*

**p5l19 I would like more details on the process to remove SPEs. Anything above a local average is removed? Would a method such as a hampel filter be more appropriate here? I would like a little more detail here.**

*SPEs were removed in two steps.*

*Step 1):*

*Based on the „Solar Proton Event Archive" (http://space-env.esa.int/index.php/Solar-Proton-Event-Archive.html) provided by NOAA SEC, SPEs were removed for all near Earth spacecraft. Following http://space-env.esa.int/index.php/NOAA_SPE_Template.html?date=19971104 :"The*

*Event selection criterion is when the NOAA/GOES-9 five minute averaged >10. MeV p+ flux exceeds 2.0 p+/cm²/s/sr. The event is considered to have ended when the flux returns to below 1 p+/cm²/s/sr". Figure 2 shows an example of SPE period rejection. Since the data is based on geostationary satellites, further SPEs detected by HEND and Rosetta at locations with a significant longitudinal difference with respect to the Earth's heliocentric longitude had to be removed as discussed in step 2).*

*Step 2):*

*Numerical outlier detection was applied onto the data sets using a rolling mean outlier detection method. Due to the very noisy nature of the data set it turned out that one would either throw away too much data or would stick with still many outliers. Therefore it was decided to remove all non reported SPEs by hand.*

[Figure]

*Figure 2: Example of SPE rejection for the INTEGRAL radiation monitor (for the referee only)*

*We have added the link to http://space-env.esa.int/index.php/NOAA_SPE_Template.html?date=19971104 in the relevant paragraph.*

**Figures 2 and 3 can you quote a value for the goodness of fit like chi^2**

*The chi2 is now included. It is equal to ~ 4 in the case of Rosetta/INTEGRAL, and ~6 in the case of HEND/Rosetta.*

**Figure 4 and the related discussion the sun spot number is displayed but there is no source for this data. There are several different metrics which can be used as a 'sunspot number' see Lockwood 2014 and refs therein https://doi.org/10.1002/2014JA019970**

*Our source was : https://spitfire.estec.esa.int/ODI/dplot_ssn.html*

*This information was added in the acknowledgements.*

**Figure 4 what is the cadence of the data in Figure 4, are these averaged with over 27 days also?**

*Yes, we added labels to the plot stating this.*

**It would be interesting to also plot the variance or the standard deviation for the same window width as the averaging of the magnetic field as a proxy for the fluctuation amplitude of the magnetic field fluctuations.**

*Work is ongoing to see the effect of magnetic fluctuations (effect of turbulence), therefore we leave this activity for the near-future.*

---

## Author Comment (AC3) · 7 Aug 2019

**Replies to E. Roussos' comments**

**1) Equation 1 is used to estimate radial gradients. However, N1 & N2 are count-rates, which are proportional to integral fluxes. Therefore, the estimated parameter is an "integral gradient". "Differential gradients" require to have differential flux measurements. For instance, it is my understanding that Gieseler & Heber (2016) estimate differential gradients, so comparison with the values obtained in this study should be reconsidered, even if values are similar.**

*Indeed, the data used in this study are counts that are proportional to integral fluxes. Differential GCR fluxes have not been yet extracted from the SREM data. Work is ongoing. In the manuscript, we now make it clear that our estimated parameter Gr is an integral gradient.*

**2) Both differential and integral gradients have an energy dependence. For the latter, which are more relevant to the present study, it matters above which energy fluxes are integrated. The used channel captures protons >49 MeV, however, from other SREM papers it seems that the geometry factor <100 MeV is rather low. So, I assume the estimated gradients have are for protons much above 100 MeV. Maybe folding the response function of the TS2 channel with a standard GCR spectrum can show which energies dominate.**

*The following plot show the SREM GCR response (the X-axis is the energy in MeV). We can see that TC2 is mostly sensitive to particles in the range [200-20000] MeV. We now indicate this range when we compare with previous results (see the updated section 3.2).*

[Figure]

*Figure 1: SREM GCR response*

**3) I am not sure how the HEND data are used in the study. In order for them to be compared with those from SREM, they have to be normalized to the INTEGRAL countrates, since SREM data are normalized to the INTEGRAL measurements. This means that in the y-axis of Fig. 3, one should used the INTEGRAL-normalized rates of SREM, not the raw SREM rates. I.e. this has to be a 2-step normalization. If that was actually done, its has to be clarified in the text.**

*This was actually done: we did a 2 step normalization: Rosetta to INTEGRAL and calibrated Rosetta to HEND. This is now clarified in the text.*

**4) After HEND data are normalized to SREM, they were not used in any part of the analysis. E.g. they may also be used to estimate radial gradients, which should be similar to those coming from the SREM/INTEGRAL ratios, otherwise they may be indicative of uncertainties in the gradient estimation, or, even better, of a radial dependence of the ratios. Instead, HEND are only mentioned briefly in lines 5-15 in p.11.**

*We found out (see the attached report, Thomas Honig's internship) that the HEND data set was not suitable for estimating the radial gradients. The data is too noisy. We used HEND in the analysis of the anticorrelation with interplanetary magnetic field and sunspot number (Annex 1). We have added more plots in this annex.*

**5) In addition to the comment above, it is clear that in the comet phase, where SREM sees a negative radial GCR gradient, the gradient between INTEGRAL/HEND is clearly positive, even if normalization may require an update (see comment 3). That further supports the possibility of a reduction of GCR fluxes around the comet. My suggestion is the following: a)Estimate the radial gradient between INTEGRAL/HEND for times during Rosetta's comet phase b)From this radial gradient, estimate what should have been the count rate of SREM c)Estimate the difference between the expected and the measured count-rate d)This difference may be estimated also by using in step (b) the average positive radial gradient as found from the data shown in Fig. 5 e) Then, the difference (estimated by any of the methods) could be organized as a function of heliocentric distance (essentially activity) or any other relevant parameter.**

*In fact, we applied this procedure (see attached report, internship report of Thomas Honig) with INTEGRAL. The steps were:*

*a) Compute a radial gradient with the INTEGRAL and Rosetta data, for the time period covering the Rosetta cruise phase.*

*b) Assuming the same gradient for the comet phase, we simulated the Rosetta SREM count rate, from the INTEGRAL data, taking into account the correct distances.*

*c) When comparing the simulated and measured SREM count rates, we find this decrease in GCR count, see plot below.*

*In order to make the article not too heavy, we did not explain this procedure, and we only included Figure 5 to illustrate the GCR "absorption". Following this comment and the referee's comments, we will now include a new figure (new figure 7):*

[Figure]

*Figure 1: New Figure 7.*

*We have updated the text regarding the fact that the gradient between INTEGRAL/HEND is clearly positive (also following referee#2's comment).*

**It appears intriguing that in Fig. 6, the count-rate difference appears to maximize around mid-2015, close to perihelion, and tends to become zero again towards the end of the mission.**

*We looked into more details about the behaviour of the GCR counts during the Rosetta comet science phase (see attached report and the new Figure 7 above), in particular to see the effect of heliocentric distance (comet activity), Rosetta-nucleus distances etc...However, we did not find anything relevant (so far).*

**I hope the authors find these comments helpful.**

*We thank Elias Roussos for these very helpful comments.*

---

## Author Comment (AC4) · 7 Aug 2019

**Overall, the paper presents interesting findings. However, a more detailed description of the physical processes leading to eg the gradient in GCR flux would improve the scientific content significantly.**

*We thank Charlotte Goetz for her feedback.*

*We plan to update the introduction in the revised article. The following the sentence "The variation of GCRs as a function of different factors (solar cycle, heliocentric distance, solar wind conditions) is an interesting topic to explore, and lead to a better understanding of the heliosphere"is replaced by:*

*"The variation in galactic cosmic rays intensity depends on different physical processes: inward diffusion in the interplanetary magnetic field, adiabatic cooling, outward convection and deceleration in the solar wind plasma, drift along the heliospheric current sheet, and interaction with magnetic structures in shocks and in interplanetary coronal mass ejections (e.g. McKibben; Potgieter, 2013; Morral 2013; Alania et al., 2014; Kozai et al. 2014; Giseler and Heber 2016). The GCR intensity is therefore varying with the solar wind velocity, the magnitude of the interplanetary magnetic field, solar activity, the heliospheric current sheet tilt angle, and the solar polarity change."*

**Figure 1a: the green and blue are hard to distinguish, maybe another color combination would be better here.**

*The figure was updated. See below.*

[Figure]

[Figure]

1. It is unclear to me why the instruments need cross-calibration. What are the technical reasons for the instruments different behaviour if they are essentially the same model? You mention sensitivity area, can you elaborate further? Do you have any reason to believe that the dependence of the countrate of two instruments is linear? Could it also be something else? (second or third order?)

*Even if the radiation monitors are of the same design, they are not identical copies. It is therefore reasonable to assume that they will have similar performances, but not equal. In fact, the purpose of the cross-calibration is:*

*a) Show that the radiation monitor family behave similarly. This is a good engineering achievement, useful to report.*

*b) The response of a particle detector also depends on the radiation environment it is exposed to. In this case, SREM detectors are mounted on/within spacecraft, which may generate secondary particles that modify the radiation environment at the detector to be different from that in deep space. Therefore, the instrument cross-calibration would help to reduce the related uncertainties.*

*c) Normalise the count rate in case quantitative cross-mission studies are performed, like in the present article.*

*In the article, we do not elaborate further regarding the linearity of the calibration. Second or higher orders are always possible, but looking at the behaviour (like see the plots below for Herschel, Planck, INTEGRAL and Proba-1 monitors), a linear fit appears to us very reasonable to consider.*

[Figure]

[Figure]

**2. p7l3: the equation given here is not consistent with what is shown in Figure 2. From figure two the relationship should actually be: Count(Integral)=1.028 x count(Rosetta) -0.127 . Then all the other calibration functions should also be checked.**

*The relationship is correct: Integral = 1.028 Rosetta – 0.12. The text was corrected.*

**3. p11l21ff: The correlation is very obvious. You say this is something that was expected and address this briefly in the annex. I think it would be better suited here and needs to be explained in more scientific detail.**

*Page 12, we add the following sentence: "This anticorrelation is due to the modulation of GCR intensity. The GCR intensity decreases when the magnetic field and the solar activity increase due to the GCR diffusion in the solar wind. This "engineering" data is a new data set that can be useful to study this modulation.", after "In addition, the expected anticorrelation between GCR and IMF and Sun spot number was calculated and the result can be found in Annex 1."*

**4. p12l11ff: Again, a physical explanation of why this gradient is expected would be good.**

*This was added. It is here only a question of heliocentric distance. GCRs propagate in the solar system, and their flux decrease with the solar distance, due to the increasing strength of the magnetic field.*

*New sentence: This positive gradient is mainly due to the inward diffusion of GCRs in an interplanetary magnetic field whose strength decreases with heliocentric distance.*

**5. p15l22: However, they ARE highly.....**
**6. p15l31: demonstrate –> demonstrated**

*Corrected.*

---

## Author Comment (AC5) · 7 Aug 2019

**Replies to referee #2**

*We thank the referee for the careful review and all these very helpful comments. Our answers are in italic blue colour, while the updated text is in red.*

**GENERAL COMMENTS**

**SREM TC2 channel is a single detector channel. Since no coincidence/veto logics are applied, this channel could include a significant contribution from secondary particles induced by cosmic ray interaction with the s/c. Authors should at least briefly discuss if this is the case, as well as the relative importance of possible sources of background. Since the study covers a long time interval and a relatively wide range of radial distances, spatial and/or temporal variations of the background can be a critical issue for the analysis of quiet-time fluxes (GCR) and could affect the reliability of cross-calibrations along the period under analysis.**

*Indeed, secondary particles can influence the measurement. Considering different mass distributions of Rosetta and INTEGRAL, it can be assumed that this influence differs for each spacecraft. Under the assumption of a continuous impact averaged over the solar period of 27 days the rather, furthermore assumed, constant contribution might be different for Rosetta and INTEGRAL, but is expected to decrease significantly when performing the cross calibration in similar space environments.*

*A detailed study on the contribution by secondary particles could be a possible part in a follow-up study, although some work was already carried out at University of Oldenburg (Validation of flux models to characterize the radiation environment in space based on current Rosetta-data, Master thesis, 2017).*

*Section 2.4 now includes: The TC2 channel could include a significant contribution from secondary particles induced by cosmic ray interaction with the spacecraft itself. As a first approximation, this contribution is expected to be minimised in the cross-calibration process. A full characterisation could be the topic of a follow-up study.*

**SPECIFIC COMMENTS**

**Page 3, line 5. Although single detector channels such as TC2 are omnidirectional, it would be helpful for the later interpretation of the data to briefly describe the pointing of the Rosetta SREM aperture during the cruise phase and during the comet encounter.**

*We agree that it would be useful to study the pointing of the SREM detector, in particular for channels like C2 who are sensitive in a ±20° cone. This should be an interesting follow-up study. Since this study mainly concerns the TC2 channel, we did not spent too much time on it. We briefly discuss this point in Thomas Honig's internship report (see attached, section 5.10). In addition, for the referee, we show here how the comet 67P looks like in the SREM field of view.*

[Figure]

*Figure 1: SREM 20x20 field of view, 1 June 2016. Courtesy A. Sanderink.*

**P3 Table 1. The second column (logic) lists "D1" as logic for channels TC2 and S25. This seems inconsistent with e.g. page 2 line 26 and with Table 1 in Evans et al., 2008. Please check/correct since channel TC2 is the basis of the study presented in the manuscript. Instead of just providing a nominal "49 MeV to infinity" energy range, it would be valuable to discuss**

**the rigidity or energy range roughly represented by TC2 counting rates (taking into account the typical shape of the modulated GCR spectrum above 49 MeV and probably the energy-dependent instrument response). This information would be quite useful when comparing the inferred radial gradient with previously published results (Page 12).**

*Agreed; there was a mistake in Table 1. The table was updated and simplified. See below:*

*Table 1: new Table 1*

| Channel | Bin | Logic | Particles | Energy range (MeV) |
|---|---|---|---|---|
| 1 | TC1 | D1 | Protons | 27-inf |
| | | | Electrons | 2-inf |
| 2 | S12 | D1 | Protons | 26-inf |
| | | | Electrons | 2.08-inf |
| 3 | S13 | D1 | Protons | 27-inf |
| | | | Electrons | 2.23-inf |
| 4 | S14 | D1 | Protons | 24-542 |
| | | | Electrons | 3.2-inf |
| 5 | S15 | D1 | Protons | 23-434 |
| | | | Electrons | 8.08-inf |
| 6 | TC2 | D2 | Protons | 49-inf |
| | | | Electrons | 2.8-inf |
| 7 | S25 | D2 | Ions | 48-270 |
| 8 | C1 | D1 x D2 | Protons | 43-86 |
| 9 | C2 | D1 x D2 | Protons | 52-278 |
| 10 | C3 | D1 x D2 | Protons | 76-450 |
| 11 | C4 | D1 x D2 | Protons | 164-inf |
| | | | Electrons | 8.1-inf |
| 12 | TC3 | D3 | Protons | 12-inf |
| | | | Electrons | 0.8-inf |
| 13 | S32 | D3 | Protons | 12-inf |
| | | | Electrons | 0.75-inf |

| 14 | S33 | D3 | Protons | 12-inf |
| | | | Electrons | 1.05-inf |
| 15 | S34 | D3 | Protons | 12-inf |
| | | | Electrons | 2.08-inf |

*We agree that it would be more useful to provide more precise number than the "nominal "49 MeV to infinity" energy range". However, this is clearly out of scope for this study. Nevertheless, for the TC2 channel, we can provide additional information. The following plot show the SREM GCR response (the X-axis is the energy in MeV). We can see that TC2 is mostly sensitive to particles in the range [200-20000] MeV. We now indicate this range when we compare with published results.*

*Added in table 1 caption: The study of the detector response to GCR indicates that the TC2 channels is mainly sensitive to energies between 200 MeV and 20 GeV.*

*New sentence in section 3.2: This result agrees well with previous studies for which the energy range can be compared with the TC2 range of ~0.2 – 20 GeV (e.g. Vos and Potgieter, 2016 / range 0.1-10 GeV; Gieseler and Heber, 2016 / range 0.45-2 GeV).*

[Figure]

*Figure 2: SREM detector response to GCR (for the referee only)*

**P4 Figure 1, bottom panel. Please define/clarify HEE, since this acronym normally stands for the Heliocentric Earth Ecliptic coordinate system and therefore latitudes would correspond to ecliptic latitudes and not to heliographic latitudes. Do the authors mean HEEQ (Heliocentric Earth Equatorial)? If not, replace "heliolatitude" by "ecliptic latitude". This difference is also relevant for the discussion in P14L10.**

*"heliolatitude" was replaced by "ecliptic latitude". Figure 1b was updated, see below.*

[Figure]

*Figure 3: New Figure 1b*

**P5L19. The SREM channels and the exact procedure used to manually filter solar energetic particle events should be specified. Some events could be difficult to detect in TC2 (but still contribute) but become clearly visible at energies below 49 MeV.**

*SPEs were removed in two steps.*

*Step 1):*
*Based on the „Solar Proton Event Archive" (http://space-env.esa.int/index.php/Solar-Proton-Event-Archive.html) provided by NOAA SEC, SPEs were removed for all near Earth spacecraft. Following http://space-env.esa.int/index.php/NOAA_SPE_Template.html?date=19971104 :"The Event selection criterion is when the NOAA/GOES-9 five minute averaged >10. MeV p+ flux*

*exceeds 2.0 p+/cm²/s/sr. The event is considered to have ended when the flux returns to below 1 p+/cm²/s/sr". Figure 2 shows an example of SPE period rejection. Since the data is based on geostationary satellites, further SPEs detected by HEND and Rosetta at locations with a significant longitudinal difference with respect to the Earth's heliocentric longitude had to be removed as discussed in step 2).*

*Step 2):*
*Numerical outlier detection was applied onto the data sets using a rolling mean outlier detection method. Due to the very noisy nature of the data set it turned out that one would either throw away too much data or would stick with still many outliers. Therefore it was decided to remove all non reported SPEs by hand.*

[Figure]

*Figure 4: Removing SPE events (for the referee only)*

*The figure illustrates exemplarily how SPEs were removed by visual inspection (by removing the gray area) and how radiation belt influences on INTEGRAL were removed (red data points were removed with requirement: distance to Earth > 60000km). The quite period as of 2005-01-25 also shows good agreement between uncalibrated Rosetta and INTEGRAL in a similar radiation environment on average.*

*We have added the link to http://space-env.esa.int/index.php/NOAA_SPE_Template.html?date=19971104 in the relevant paragraph.*

**P5L25. The reconfiguration of HEND should be explained with further detail. What is the ultimate reason for the count rate increase/offset?**

*In the HEND data set, there is the errata.txt file (http://pds-geosciences.wustl.edu/ody/ody-m-grs-2-edr-v1/odge1_xxxx/errata.txt) which report about this reconfiguration. This is the only public information that we found. The updated manuscript provides now this reference, in addition to [J. J. Plaut, personal communication]*

**P6L5-6. "We associate the 2.8% difference….with differences in the sensitivity area of the two SREM detectors". Other factors such as noise levels, obstructions or different s/c mass distribution around the sensor head could contribute to this small difference between the nominally identical units onboard INTEGRAL and Rosetta (as well as Herschel, Planck and Proba-1).**

*Agreed; the updated manuscript now lists the other factors as well.*

**P7L23. Is this equation correct? The preceding text mentions that Proba-1 counting rates are systematically lower than INTEGRAL counting rates, then why is INTEGRAL rate (and not Proba-1 rate) multiplied by a number >1 in order to obtain a "corrected" Proba-1 rate directly comparable with INTEGRAL? The same question applies to the rest of cross-calibration equations. Probably this is just a notation problem, but it should be checked.**

*We have double checked and the cross calibration functions have to be changed as follows:*

*Count(INTEGRAL) = 1.028 x Count(Rosetta) – 0.127 / s*
*Count(INTEGRAL)  = 0.931 x Count(Herschel) + 0.060 / s*
*Count(INTEGRAL)  = 0.938 x Count(Planck) + 0.028 / s*
*Count(INTEGRAL)  = 1.256 x Count(Proba-1) + 0.154 / s*
*Count(Rosetta)  = 0.035 x Count(HEND) – 0.557 / s*

**P8L6. See comment P7L23 above and clarify how the calibration factors are applied to HEND data in order to make them directly comparable with Rosetta.**

*We did a 2-step normalization: Rosetta to INTEGRAL and then calibrated Rosetta to HEND.*

*The text now says: The HEND neutron monitor HEND is calibrated with respect to SREM-Rosetta, which is calibrated with respect to INTEGRAL*

**P10 Figure 4. What is the reason for the peaks observed in HEND rates shortly after the vertical red dashed line? The difference between the near-Mars HEND and near- Earth SREM rates seem larger at the end of the plot (2015-16) than at the beginning (2003), while solar activity levels are comparable (although with opposite solar polarity). This, together with the comment on P5L25 raises some doubts about the stability of cross-calibration. Some discussion of these differences could be introduced e.g. in P11L7-8. Visibility of the different lines in the first panel could be improved.**

*Figure 4 was updated, see below:*

[Figure]

*The HEND spikes after the vertical red lines were removed, as part of the SPE removals. We should have done that for the submitted manuscript. The larger differences between HEND and INTEGRAL in 2015-2016 are not understood, and are now outlined in the manuscript.*

**P12L13-14. When citing agreement with Vos and Potgieter, 2016 and Gieseler and Heber, 2016, please mention the rigidity/energy ranges studied by these authors. The radial gradients presented here are based on counting rates accumulated over a broad energy**

**range, which makes difficult to define a reference energy for comparison (see comment about Table 1 above).**

*The text was updated accordingly: … (e.g. Vos and Potgieter, 2016 / range 0.1-10 GeV; Gieseler and Heber, 2016 / range 0.45-2 GeV)*

**P12L16-17 and P14 Figure 6. "After that period, the count rate variation is in very good agreement with the expectation of a positive(?) radial gradient…". This sentence sounds quite confusing since the Rosetta rates remain below the INTEGRAL rates till late 2016 (Figures 4 and 5 and discussion in following pages). In order to make easier the interpretation of Figure 6, I strongly suggest including panels showing the heliocentric distance of Rosetta and the distance between Rosetta and Comet 64P. In order to keep consistency with Figure 4, I also suggest plotting INTEGRAL SREM rates in green color.**

*There are three periods of interest:*
- *Blue dots and red fit: Rosetta pre-hibernation data. Here we see the expected positive gradient, 2.96% per AU.*
- *Orange dots are January-July 2014: period when the attenuation of GCRs is noticed.*
- *Green dots and black fit: July 2014-September 2016. The attenuation is stable, and we see again the expected positive gradient, 2.9% per AU.*

*The text now says: „After that period (green points and black fit), the count rate variation and the ratio is in very good agreement..."*

*Figure 6 was updated and made colors consistent. A new Figure was included (Figure 7) which displays the relevant distances. See below.*

[Figure]

*Figure 5: New Figure 6*

[Figure]

*Figure 6: new Figure (figure 7), which includes the heliocentric distance of Rosetta and the Rosetta-comet distance.*

**P14L13. "The decreasing ratio begins when Rosetta reaches around 20,000 km from the cometary nucleus". See comment above about including distance between Rosetta and the comet in Figure 6.**

*Figure 7 now includes the Rosetta-nucleus distance.*

**P14 and P15. Please briefly mention that the counting rates at Mars/HEND always stay higher than those registered near the Earth (Figure 4), even during the period shown in Figure 6. This is consistent with a permanent positive GCR radial gradient and supports that the reduction in the GCR rates at Rosetta compared to Earth is related to the comet approach.**

*The text was updated with the suggested sentences.*

**P15L16-17. In order to substantiate this suggestion, the authors should consider including and discussing the local plasma and magnetic field observations by Rosetta during the period shown in Figure 6.**

*Work is ongoing to see the effect of magnetic fluctuations (effect of turbulence); therefore, we leave this activity for the near future.*

**MINOR AND TYPOGRAPHIC COMMENTS**

**Affiliations: Please replace "Universitycity" by "University".**

*Corrected.*

**Page 1, Line 21. "Major sources of this radiation are the Van Allen radiation belts,: : :".**
**The preceding sentences put the focus on missions outside the Earth's magnetosphere,**
**therefore this reference to radiation belts seems out of place.**

*Agreed. This reference was removed.*

**P2L4. "cover a range of heliocentric distances up to 3.5 AU". Since Rosetta reached**
**4.5 AU, the term "heliocentric distance" sounds confusing here. Do the authors refer**
**to the difference between the radial distances of two s/c, rather than to the Sun-s/c**
**distance?**

*We understand the confusion. The word "heliocentric" was removed.*

**P2L9. "with two of them still operating: : :". Please specify which ones.**

*Corrected.*

**P2L10. I suggest replacing "high energetic charged particles" by "high-energy charged**
**particles" here and elsewhere in the manuscript.**

*Corrected in three places.*

**P3 Table 1. I suggest avoiding the redundant "MeV" labelling in the first and second**
**line of the header, e.g. keeping it only in the first line.**

*Table 1 was updated and simplified, see above.*

**P3L21. "This sensor is the best one for space weather: : :". Please reformulate this**
**sentence in a more specific way.**

*"the best one" was changed by "very adequate".*

**P4L11. Replace "Figure 1 shows: : :." by "Top panel of Figure 1 shows: : :".**

*Corrected.*

**P5L5. Please, replace "and HEND's orbits: : :" by "and Mars Odyssey's orbits: : :".**

*Corrected.*

**P5L15. What is the energy and/or intensity threshold of the Solar Proton Event list used**
**here?**

*We refer here to an answer above.*

**P6L3. I suggest replacing "the fit curve" by "the linear fit".**

*Corrected.*

**P7L20. Since the energy threshold of SREM TC2 is relatively low (49 MeV), magnetospheric shielding of the lower energy part of the GCR spectrum could also play a role here.**

*We used data above the magnetosphere (height above 60,000 km); therefore, we think that the magnetosphere has little effect.*

**P7L39. I suggest replacing "The neutron monitor HEND" by "The HEND neutron detector" or just "HEND- Mars Odyssey".**

*Corrected.*

**P8L8. Indeed, the possible effect of Mars shadow would be implicitly included in the empirical cross-calibration procedure.**

*Since we did not do anything special, the Mars shadow is implicitly included in the cross-calibration.*

**P8 Table 2. Please add the units of b coefficient in the first row and include a and b uncertainties in the table.**

*The table was updated accordingly:*

| Spacecraft 1 | Spacecraft 2 | a | $\Delta a$ | b [1/s] | $\Delta b$ [1/s] |
|---|---|---|---|---|---|
| INTEGRAL | Rosetta | 1.028 | 0.005 | -0.127 | 0.017 |
| INTEGRAL | Herschel | 0.931 | 0.001 | 0.060 | 0.005 |
| INTEGRAL | Planck | 0.938 | 0.001 | 0.028 | 0.005 |
| INTEGRAL | PROBA-1 | 1.256 | 0.002 | 0.154 | 0.005 |
| Rosetta | HEND | 0.035 | 0.002 | -0.557 | 0.025 |

**P10 Figure 4 caption, P11L10 and P11L27. Replace "sun spot" by "sunspot".**

*Corrected.*

**P11L5 and P15L25 "solar rotation (27 days)" While this is OK for the purpose of smoothing longitudinal and transient variations at all locations, 27 days is just the Carrington (synodic and at intermediate latitude) solar rotation period. Therefore, I suggest removing "one solar rotation" and leaving just "27 days".**

*Corrected.*

**P11L9 "different heliographic locations". Do the authors mean "heliospheric locations"?**

*Corrected.*

**P11L12 "one of its orbits".**

*Corrected.*

**P11L20 "long-term".**

*Corrected.*

**P11L27 "anticorrelation: : :was calculated". I suggest replacing either by "anticorrelation: : :was analyzed" or by "correlation coefficient: : :.was calculated".**

*Corrected.*

**P12L14 and P13L10 the term "comet phase" should be clarified/defined.**

*The comet phase refers to when Rosetta started its nominal mission around comet 67P. This is now clarified.*

*New text:* "(the start of this phase is marked by the red vertical bar on Figure 1b)" *was added after "The slope during the comet phase".*

**P13L12. Replace "dark" by "black".**

*Corrected.*

**P14L16 "hundreds of kg of propellant: : :etc.". Could this mass loss significantly change the mass environment around the SREM detector and reduce the rate of secondaries contributing to TC2? For completeness, the authors could also mention the separation date of the Philae module and its (small) mass.**

*We checked during our study that the mass loss due to manoeuvres was not correlated with the SREM data variations.*

*We added the sentence "For completeness, we checked that the separation with the Philae module in November 2014 did not have noticeable effect."*

**P15L22 "they are highly: : :".**

*Corrected.*

**P15L29 "sunspot".**

*Corrected.*

**P15L29. Please replace "annex" by "Annex 1".**

*Corrected.*

**P15L30 Please replace "geographically" by "spatially".**

*Corrected.*

**P15L31 "demonstrated".**

*Corrected.*

**P16L3. The period corresponding to this radial gradient should be mentioned here.**

*"(between, 2004-10-21 to 2011-05-21)" was added.*

**P16L4 "this information provides".**

*Corrected.*

**P18L27 Replace "cosmis" by "cosmic".**

*Corrected.*

**P19L24 Replace "intergral" by "INTEGRAL".**

*Corrected.*

**P22 Figure Annex 2. A panel showing the azimuthal separation between Rosetta and the Earth could be added to illustrate the origin of the time delays in the observed o-rotating structures.**

*Agreed;the azimuthal separation was added in the figure, see below. The caption was updated accordingly.*

[Figure]

*Figure 7: New Figure of Annex 2*

*The first panel shows the azimuthal separation between Rosetta and INTEGRAL. The second panel shows the count rates of channel TC2 as before. Both panels share the same x-axis (below first panel and above second panel). We see a nice (and expected) correlation between Rosetta and INTEGRAL approaching each other in terms of azimuthal separation and a decreasing time shift in the measured count rates until the point when delta phi vanishes and the count rates match well. Furthermore, color for INTEGRAL has been made consistent.*